# Feedback for Emergency Ambulance Staff: A National Review of Current Practice Informed by Realist Evaluation Methodology

**DOI:** 10.3390/healthcare11162229

**Published:** 2023-08-08

**Authors:** Caitlin Wilson, Gillian Janes, Rebecca Lawton, Jonathan Benn

**Affiliations:** 1School of Psychology, University of Leeds, Leeds LS2 9JT, UK; 2Research and Development Department, Yorkshire Ambulance Service NHS Trust, Wakefield WF2 0XQ, UK; 3Yorkshire Quality and Safety Research Group, Bradford Institute for Health Research, Bradford BD9 6RJ, UK; 4Faculty of Health and Education, Manchester Metropolitan University, Manchester M15 6BH, UK

**Keywords:** ambulances, Emergency Medical Services, feedback, health services evaluation, organizational case studies

## Abstract

Research suggests that feedback in Emergency Medical Services (EMS) positively affects quality of care and professional development. However, the mechanisms by which feedback achieves its effects still need to be better understood across healthcare settings. This study aimed to understand how United Kingdom (UK) ambulance services provide feedback for EMS professionals and develop a programme theory of how feedback works within EMS, using a mixed-methods, realist evaluation framework. A national cross-sectional survey was conducted to identify feedback initiatives in UK ambulance services, followed by four in-depth case studies involving qualitative interviews and documentary analysis. We used qualitative content analysis and descriptive statistics to analyse survey responses from 40 prehospital feedback initiatives, alongside retroductive analysis of 17 interviews and six documents from case study sites. Feedback initiatives mainly provided individual patient outcome feedback through “*pull*” initiatives triggered by staff requests. Challenges related to information governance were identified. Our programme theory of feedback to EMS professionals encompassed context (healthcare professional and organisational characteristics), mechanisms (feedback and implementation characteristics, psychological reasoning) and outcomes (implementation, staff and service outcomes). This study suggests that most UK ambulance services use a range of feedback initiatives and provides 24 empirically based testable hypotheses for future research.

## 1. Introduction

Feedback on clinical performance is well researched, and systematic reviews suggest it results in small to moderate improvements in patient care by enhancing healthcare professionals’ compliance with desired clinical practice across healthcare settings, including Emergency Medical Services (EMS) [1,2]. EMS professionals work autonomously in an environment characterised by complexity, uncertainty and extreme stressors [3,4]. They make complex decisions, including assessing and treating patients at home to avoid unnecessary hospital attendance and reduce demand [5,6]. For EMS staff, receiving feedback on clinical practice and decision-making is vital, yet often difficult due to practical constraints in obtaining information concerning patients not conveyed to hospital and barriers to information transfer across the boundary marked by hospital admission.

Research from North America and the United Kingdom (UK) suggests that EMS staff view current feedback provision as inadequate and desire more feedback, especially concerning patient outcomes [7,8,9]. Patient outcome feedback involves providing clinicians with information regarding what happened to their patients to support reflection and self-evaluation of clinical performance [10]. Meanwhile, clinical performance feedback predominantly involves auditing clinical performance over time and feeding this back to clinicians in summary form (‘*audit and feedback*’) [11].

A recent meta-analysis supported a moderate positive effect for various feedback types within EMS, including clinical performance and patient outcome feedback [2]. The review highlighted a need for more theoretical underpinning and robust evaluation of existing feedback in EMS regarding outcomes and understanding the mechanisms by which feedback operates within this context [2].

The broader literature on clinical performance feedback suggests that while feedback overall has small to moderate positive effects, these effects vary greatly depending on the context, initial performance and mode of feedback provision [1]. Despite researchers attempting to optimise the effects of feedback across healthcare settings [12,13,14], key active components and mechanisms by which feedback works still need to be better understood [15]. The most recent theory development within audit and feedback in healthcare is Clinical Performance Feedback Intervention Theory (CP-FIT), based on a meta-synthesis of 73 feedback interventions [16]. Although none of the 73 interventions were within EMS, our provisional logic model developed from qualitative work in one ambulance trust found CP-FIT to have good face validity when exploring how feedback impacts patient care, patient safety and staff wellbeing [9].

Examples of clinical performance and patient outcome feedback within EMS in the UK include paramedics receiving feedback on cardiac arrests [17] and accessing post-box schemes at emergency departments to directly follow up on patients [18] or request facilitator-mediated feedback [19]. Considerable variation exists between these initiatives, which appear to be primarily isolated schemes developed to meet local priorities and needs without systematic evaluation [2]. This may prevent collective learning and increases the risk of individual feedback initiatives failing [20].

This study aimed to understand how UK ambulance services provide feedback to their staff and develop a programme theory of how feedback works within EMS. Programme theories make sense of complex interventions by drawing out the underlying principles of how an intervention causes outcomes within a specific context [21,22,23]. The following questions were addressed:What are the types and design elements of interventions providing feedback for EMS professionals in the UK?What is it about EMS feedback interventions that works, for whom, in what circumstances, how and to what extent?

## 2. Materials and Methods

### 2.1. Study Design

This was a mixed-methods study informed by realist evaluation methodology. Realist approaches seek to explain underlying mechanisms of action and encourage using substantive theory from other disciplines, for example, where research into feedback is more advanced [24,25]. The study consisted of two work packages (WPs): a cross-sectional survey of EMS feedback initiatives in the UK (WP1) followed by purposively sampled case studies involving documentary analysis and semi-structured interviews (WP2) (Figure 1).

The study was carried out in accordance with the UK Policy Framework for Health and Social Care Research [26]. It was approved by the University of Leeds ethics committee (PSYC-242 7 July 2021) and the Health Research Authority (IRAS project ID 287110). The RAMESES II reporting standards for realist evaluations [27] and Checklist for Reporting Of Survey Studies [28] guided study reporting.

### 2.2. Setting and Participants

The study setting was UK emergency ambulance services. Eligible participants for WP1 were leads of EMS feedback initiatives, defined as “*providing feedback to emergency ambulance staff regarding their performance within prehospital practice and/or patient outcomes*”. We used a sampling framework to purposively select four case studies from the feedback initiatives identified in WP1. The sampling framework was stratified according to contextual and feedback intervention characteristics, including whether feedback was actively sought out by EMS staff (‘*pull feedback*’) or provided without solicitation (‘*push feedback*’). Eligible participants for WP2 were initiative leads and emergency ambulance staff who had received feedback through a case study initiative.

### 2.3. Data Collection

The survey and interview guides (Appendix A) were developed by an early career paramedic researcher (C.W.) with input from a senior health services researcher with experience in evaluating feedback initiatives (J.B.). The survey for WP1 was hosted on Qualtrics (Qualtrics, Provo, UT) from September 2021 to January 2022 and contained six sections: respondent demographics, feedback initiative description, mechanisms, implementation, consequences and unique characteristics. The questions were open- and closed-format and included qualitative (e.g., project aims, barriers) and quantitative (e.g., start date, numbers of requests) items.

No formal sample size calculation was conducted as this was an exploratory survey aiming to capture all eligible EMS feedback initiatives. We used purposive and snowball sampling to send the survey to programme leads identified through grey literature, networking and clinical practice. The survey was also advertised on social media and via the research departments of participating UK ambulance services. Informed consent was obtained at the beginning of the survey, which automatically ended if users opted not to consent. The anonymity of individual participants was maintained.

To begin WP2, initiative leads were invited to participate in a semi-structured interview and provide existing documents linked to their initiative using the contact details provided in WP1. A further 2–4 interviewees within each case study were sampled using snowball sampling. C.W. conducted the interviews online from November 2021 to August 2022 and transcribed them verbatim. Prior to each interview, C.W. obtained written informed consent in electronic format. In line with realist evaluation methodology, interviews and documents were used to inspire, validate, falsify and modify hypotheses about how the intervention works [29]. During interviews, this was achieved by C.W. describing an aspect of the initial programme theory and allowing participants to explain and clarify based on their perspectives [30].

### 2.4. Data Analysis

Data analysis was conducted by an early career paramedic-researcher (C.W.) with input from the wider research team of senior health services researchers and behavioural scientists (G.J., R.L., J.B.). Qualitative survey items were analysed using content analysis supported by NVivo (Version 12 Plus, QSR International, Burlington, MA, USA). Quantitative survey items were analysed using descriptive statistics. Descriptive statistics were presented as frequencies (percentages) and median (interquartile ranges (IQR)). Free-text responses were categorised by C.W., with 25% (n = 10) validated by J.B.

### 2.5. Development of Programme Theory

The initial programme theory was developed by triangulating a logic model developed by the research team using abductive analysis in a prior interview study [9] with the results from WP1 and mapping this onto a Context–Mechanism–Outcome configuration (CMOc) in line with realist methodology [24]. These configurations set out the causal links between the specific mechanisms that trigger intervention outcomes within particular contexts [31,32]. Context here pertains to the background and implementation setting, while mechanisms describe how resources bring about change and outcomes relate to intended and unintended consequences [32,33,34]. We followed the CMOc approach by Dalkin et al. [34], dividing ‘mechanisms’ into ‘reasoning’ and ‘resources’ to capture how an intervention brings about change:*Context*: What are the barriers and facilitators to implementing feedback in EMS?*Mechanism—Resources*: What are the main characteristics of the type of feedback being provided?*Mechanism—Reasoning*: What are the psychological responses by which feedback achieves its effects?*Outcome*: What are the EMS feedback interventions’ perceived effects and impact?

The programme theory was refined by analysing the interviews and documents in WP2 using a retroductive approach that uses inductive and deductive reasoning—including researcher insights—to theorise hidden mechanisms [35,36,37]. In addition, we drew upon a systematic review of feedback in EMS [2], as well as existing middle-range theories from implementation science (CP-FIT [16], implementation outcomes taxonomy [38]) and behaviour change theory (mechanisms of action [39]).

## 3. Results

The survey yielded 46 responses, of which we excluded duplicate responses (n = 2) and incomplete questionnaires (n = 4), where participants entered their details but did not answer feedback intervention questions. Therefore, 40 unique EMS feedback initiatives were included in the analysis.

### 3.1. Characteristics of Survey Participants

Participants completing the survey were mainly based in emergency departments (14 consultants, 2 advanced clinical practitioners) or ambulance services (n = 21), often with managerial or specialist clinical responsibilities. The remaining participants were specialist healthcare professionals based in hospitals (n = 2) or regional networks (n = 1). Responses were from England (n = 27), Scotland (n = 8), Wales (n = 4) and Northern Ireland (n = 1), and from within 11 of the 14 UK ambulance services.

### 3.2. Types and Design Elements of Interventions Providing Feedback for EMS Professionals

Table 1 summarises the characteristics of the 40 EMS feedback initiatives. Feedback initiatives most frequently provided patient outcome feedback (n = 31, 77.5%) and less frequently clinical performance feedback (n = 4, 10.0%) or a combination of the two (n = 5, 12.5%). Patient outcome feedback included information on either the hospital diagnosis, thereby allowing paramedics to confirm their prehospital diagnosis (n = 36, 90%); patients’ hospital treatment, investigations and care trajectory to inform paramedics’ future clinical reasoning (n = 25, 62.5%); or answers to specific queries (n = 9, 22.5%). Clinical performance feedback included information on paramedics’ decision-making, such as appropriateness of conveyance and destination (n = 7, 17.5%) or compliance with EMS protocols and key performance indicators (n = 5, 12.5%), e.g., on-scene times and adherence to checklists.

Of the initiatives, 70.0% (n = 28) were active when the survey was completed, with the remaining 12 initiatives (30.0%) either permanently terminated or paused temporarily due to the COVID-19 pandemic or other service pressures. Thirty-eight respondents provided a start date for their initiative, with 77.5% (n = 31) set up in the last five years. The earliest start date was 2004; however, this initiative was no longer active. Of the currently active initiatives, the earliest was set up in 2011.

Initiatives had most commonly delivered 10–49 (n = 14, 35.0%) or 50–99 instances of feedback (n = 11, 27.5%) and predominantly provided feedback to frontline EMS staff (n = 36, 90.0%) at an individual level (n = 32, 80.0%). Email or an electronic document was the most frequently used format (n = 35, 87.5%), with provision usually occurring on an ad hoc basis (n = 29, 72.5%). The median lag time was 21 days (IQR 7-30). Feedback was most commonly for individual patient cases (n = 38, 95.0%) using a ‘*pull*’ model (n = 25, 62.5%), which involved paramedics actively requesting feedback. Initiatives were usually not part of a broader organisational or educational initiative (n = 34, 85.0%), did not include an action plan (n = 28, 70.0%) and were not underpinned by existing theory (n = 24, 60.0%).

The most frequently stated initiative aims were to close the prehospital feedback loop (n = 19, 47.5%) and enhance EMS professionals’ ability to reflect and learn (n = 21, 52.5%). The main barriers surrounding initiative development were information governance (e.g., patient confidentiality, data protection, data sharing agreements, Caldicott guardians, confidentiality advisory group approval; n = 18, 45.0%) and technology (e.g., lack of data linkage, poor data quality, data security limitations; n = 7, 17.5%). Ongoing concerns related to initiatives not being sustainable due to the time and resources required to generate the feedback (n = 16, 40.0%). The main facilitators were collaborative working and support from hospitals and broader networks (n = 14, 35.0%), the enthusiasm of EMS professionals to receive feedback (n = 9, 22.5%) and engagement from hospital staff (n = 8, 20.0%), which one initiative supported by accrediting involvement as part of continuing professional development portfolios.

### 3.3. Characteristics of Case Study Sites

The case study sites included two patient outcome feedback initiatives, one clinical performance initiative and one encompassing both patient outcome and clinical performance feedback (Table 2). They were a mixture of ‘*push*’ and ‘*pull*’ initiatives from across the UK. Seventeen interviews and six documents were analysed. The interviews lasted 29–93 min, with a median duration of 45 min. Most of the 17 participants were male (n = 11, 64.7%), and the median participant length in service was 14 years, ranging from 4 to 38 years. Participants were EMS managers or specialist clinicians (n = 7), regional network clinicians (n = 1), paramedics (n = 8) and emergency medical technicians (n = 1).

#### 3.3.1. Case Study 1

This initiative was set up at a rural hospital where staff turnover was high and ambulance handover delays were long. EMS staff “*did not typically feel like they could go and ask [hospital staff] for feedback*” (CS1-P1) because they “*did not want to bother people*” (CS1-P2), and there was no existing formal feedback mechanism. Local frontline staff generated the idea for this feedback initiative and co-designed it with the quality improvement team, including the format of requesting feedback (paper form) and receiving the follow-up information (Email). Co-design resulted in a sense of ownership (“*we said, we did*” CS1-P1). Participants looked after the physical post-box and understood how the initiative worked, so fewer instructions were required. The initiative “*successfully created a process for ambulance staff to obtain feedback locally*” (CS1-D1) but was terminated after six months due to not being sustainable because it relied on the initiative lead manually retrieving the request forms, making contact with hospital clinicians and Emailing the feedback to the requester.

Staff receiving the feedback were “*very satisfied*” or “*satisfied*” (CS1-D2) with the feedback process and were optimistic about its format and timeliness, including that it fit well with their current way of working: “*If you have a patient you are interested in, just fill out these details and then there’s a post-box in the drugs cupboard—literally just stick it in there*” (CS1-P3).

Feedback recipients indicated that the feedback positively impacted their reflective or clinical practice as they “*normally never get closure*” (CS1-D1). Participants described how the feedback had increased their confidence by allowing them to “*hear that you’ve been right about something, especially when you’re not 100% sure*” (CS1-P3) and provided them with closure as it “*puts your mind at rest*” (CS1-P4). It also changed their behaviour during subsequent patient encounters, with one participant describing that it “*informed my practice further if I saw a patient in a similar situation again*” (CS1-P2) and another recalling a specific patient encounter: “*I was very glad to have taken her to hospital and will always remember to consider pulmonary embolism in back pain in future*” (CS1-D2).

Feedback recipients expressed concerns that some staff may not engage with a feedback initiative due to viewing feedback within the ambulance service as punitive. Others might over-use it if it was rolled out at scale. However, the initiative lead’s impression was that “*people were quite conservative with it*” (CS1-P1) and talked about users “*rationing*” their requests (CS1-P2).

#### 3.3.2. Case Study 2

A condition-specific regional network set up this ‘breach-reporting’ initiative to provide feedback when EMS staff deviated from their protocol stipulating direct transport to a specialist unit for a subset of patients. Terminology for this initiative varied, whereby “*the hospital call it breach reporting […], but we [in the ambulance service] call it feedback because the primary thing that we want to do with the feedback is not to berate the crew […] but take some learning from it to improve the next time they go to a job*” (CS2-P2).

The initiative involved hospital clinicians feeding back information on inappropriate conveyance decisions to their specialist unit to a dedicated senior ambulance clinician. The senior ambulance clinician would then “*review the incident as lead clinician and Email it out to the local ambulance team leaders for face-to-face feedback to the crews*” (CS2-P2), where their decision-making was discussed to “*allow them to learn from their experience and make sure a similar patient in the next situation received the correct care that the pathway states*” (CS2-P1).

The senior clinician also aggregated feedback reports and took any “*patterns back to the learning forum to share wider*” (CS2-P2) and “*put some education packages together*” (CS2-P1). Engagement from hospital staff was achieved by introducing the new breach-reporting system during the “*staff huddle*”, ensuring the reporting did not take long to complete and identifying “champions” who sent the breach-reporting Email (CS2-P1). Engagement with senior clinicians and managers from the ambulance service was ensured early in the initiative’s development. However, no frontline staff were involved, justified by team leaders facilitating the feedback face-to-face to avoid “*demoralising*” frontline staff by sending them an Email (CS2-P1).

Initiative leads noted that since implementation several years ago, the number of breach reports had decreased, potentially indicating that the appropriateness of paramedics’ conveyance decisions had improved. However, initiative leads reported that hospital staff told them they were “*too busy at times nowadays to fill these [breach-reporting Emails] in*” (CS2-P1), which could also be the cause for decreased breach reporting. A further explanation could be increased informal feedback conversations due to improved relationships between hospital and EMS staff resulting in hospital staff being “*quite accepting in terms of asking why maybe the [ambulance] crew have done that [i.e., inappropriately conveyed a patient] rather than telling them they have done wrong*” (CS2-P1).

This initiative focused predominantly on learning; no participants mentioned staff wellbeing or closure as considerations.

#### 3.3.3. Case Study 3

This initiative used an electronic dashboard to provide patient outcome feedback to EMS professionals. A local specialist paramedic set it up at two hospital trusts using local information-sharing agreements to allow EMS staff to “*audit the quality of their work*” by displaying the paramedic’s “*provisional [prehospital] diagnosis and then comparing this with the final [hospital] diagnosis*” (CS3-D1).

Although this initiative involved information being pushed into the dashboard, access was only granted to EMS professionals who requested this from the initiative lead, who therefore classed it as a ‘*pull*’ initiative. Feedback recipients described that it “*prompted them to reflect*”, undertake “*further reading to refresh and broaden their knowledge*”, and led to “*clinical discussions with colleagues*” (CS3-D3), thereby improving “*decision-making confidence and self-efficacy*” (CS3-D2). Receiving feedback was considered a novel experience that assisted with “*pattern recognition*”, with a paramedic explaining that “*if I went to a similar job again, I would think I wonder if this is what it is and I wouldn’t have had that knowledge had I not looked it up*” (CS3-P3).

Feedback was considered particularly useful “if the ED diagnosis differed from their prehospital clinical impressions” (CS3-D3) and for patients “that were treatable in the community” where “with the right knowledge and reason, you might leave somebody at home rather than take them to hospital” (CS3-P5).

One participant compared this feedback initiative to working in primary care or hospital trusts where patient information can be accessed in a central location. Another participant thought that the rural setting of this formal feedback initiative may have contributed to the high engagement from EMS staff, as there were limited opportunities for informal feedback provision (i.e., paramedics not returning to the same hospital later in their shift and missing out on asking hospital staff about their patient’s outcome). Initiative leads reported that technological problems meant the initiative had been paused for over a year when the interviews were conducted.

When prompted, participants reported that the feedback “*provides an opportunity to clear up mysteries and support wellbeing*” (CS3-D2). Participants also noted that receiving confirmation of their prehospital clinical impression improved confidence (“*almost like a pat on the back*” CS3-P4) and aided job satisfaction by making feedback recipients “*feel more valued and more appreciated because somebody was investing time in you*” (CS3-P3). However, the main aim of this initiative was to support reflection, learning and professional development.

#### 3.3.4. Case Study 4

This initiative combined patient outcome feedback with clinical performance feedback. A senior ambulance clinician set the initiative up following an audit that indicated poor compliance in performing a specific prehospital skill, with the aim of “*sharing outcomes of what happened to patients, so the individual EMS clinicians can learn by closing the learning loop as well as have a positive impact on the wellbeing of staff because it provides a bit of closure for them*” (CS4-P1).

The initiative lead noted that gaps identified as part of this audit and feedback initiative led to an “*organisational change in how we teach and…guide our staff to manage that clinical scenario*” (CS4-P1), including a checklist to support EMS clinicians. The feedback included how compliant recipients were with the checklist, alongside patient outcomes and comments from hospital clinicians regarding how well the prehospital skill was carried out. The relevance of hospital clinicians’ comments varied with the initiative lead noting that “*prehospital care is a specialist area that hospital clinicians will not necessarily have any experience of*” (PS4-P1) and, therefore, advice may not be in line with paramedics’ scope of practice or EMS guidelines.

The lag time of this initiative was long (1–6 months), with the initiative lead concerned that providing feedback so long after the event may be “*a waste of time*” (PS4-P1) and all four feedback recipients noting they had already informally followed up on patients. However, the particular skill in question was rarely performed, meaning that all feedback recipients recalled the patient in question and that the written feedback contained “*a lot more useful information about specific injuries [that] gives you a more comprehensive understanding of what happened*” (CS4-P2). Participants expressed that they felt appreciated and that receiving feedback via this initiative boosted morale because it went above and beyond what they usually received.

Participants reported that they “*didn’t really even know there was a feedback initiative going on*” (CS4-P3) and that receiving the feedback Email “*was a bit of a surprise*” (CS4-P2). They described that the “*gentle and respectful*” (CS4-P2) tone of the written feedback meant they engaged positively despite the initial surprise. All participants discussed the feedback with their crew mates and “*reflected on how we handled the situation*” (CS4-P2). Several participants mentioned that the feedback “*made me more confident for the next time*” (PS4-P5) and had changed their practice, for example, increasing the detail of their documentation. In contrast, others stated that they kept their practice the same as they had sought feedback informally. Participants described themselves as having a positive attitude towards feedback and gave examples of when they “*provided feedback to ambulance colleagues and students, as well as receiving it*” (PS4-P4). They hypothesised that people with a less positive outlook on feedback working in an organisation that “*hasn’t got a particularly well-established culture of feedback*” (CS4-P4) might “*find it challenging*” (CS4-P2) to receive written feedback auditing their clinical performance.

### 3.4. Programme Theory of Feedback within EMS

Drawing upon analysis of the case studies, we developed 24 CMOCs, which explain how EMS feedback should work and what factors might influence its implementation and effectiveness. The CMOCs were divided into those relating to the implementation of EMS feedback interventions (Table 3, n = 11) and those relating to the effectiveness of patient outcome (Table 4, n = 9) and clinical performance feedback (Table 5, n = 4). The overall programme theory encompassed feedback and implementation characteristics (mechanism—resources), healthcare professional and organisational characteristics (context), processing and actioning feedback (mechanism—reasoning) and implementation, staff and service outcomes (outcomes), as visualised in our logic model (Figure 2).

## 4. Discussion

This study sought to draw upon multiple data sources to provide a comprehensive and theoretically informed review of practice in the under-researched area of EMS feedback. We received survey responses from 40 initiatives providing feedback to EMS professionals across the majority of UK ambulance services (11 out of 14). Most initiatives provided individual patient outcome feedback to individual frontline EMS staff using an Email format triggered by a request for feedback. Drawing upon synthesis from multiple data sources across the review of practice study and prior theory, we developed 24 CMOCs describing the mechanisms by which feedback in EMS operates to underpin our programme theory. The programme theory explains how the characteristics and implementation of an EMS feedback intervention (mechanism-resources) situated within particular healthcare professional and organisational characteristics (context) trigger individuals to process and action the feedback (mechanism-reasoning), resulting in implementation, staff and service outcomes (outcomes).

The survey revealed that EMS feedback initiatives in the UK focused predominantly on patient outcome feedback (n = 31). This contrasts with the published evidence synthesised in our systematic review [2], where most interventions focused on clinical performance feedback. The definition of feedback was the same for both studies (‘an initiative providing feedback to emergency ambulance staff regarding their performance within prehospital practice and/or patient outcomes’). Publication bias may explain the difference between what is done in practice and what gets reported in the peer-reviewed literature. Another explanation could be that EMS professionals more frequently associate ‘feedback’ with patient outcome rather than clinical performance feedback. This is consistent with findings from our interview study [9], where patient outcome feedback was the feedback type most frequently mentioned.

Patient outcome feedback was predominantly provided via pull-feedback initiatives, possibly due to this information being more challenging to obtain routinely within EMS. Although only CS3 used an electronic dashboard to provide patient outcome feedback, considerable opportunity exists to enhance feedback provision to EMS personnel through data linkage and integrated datasets spanning service boundaries [40,41]. Increased electronic data capture and feedback provision may address the lack of sustainability caused by resource constraints in generating feedback within EMS. However, careful consideration should be given to the support offered to EMS staff receiving feedback through automated electronic initiatives.

Around half of the surveyed EMS feedback initiatives reported challenges with information governance, echoing previous qualitative work [9,42]. Data governance issues related to patient outcomes being shared with clinicians that were no longer involved in a patient’s care [42], with EMS feedback initiatives taking various approaches to overcome this issue. The framing of feedback as an essential part of the learning cycle and agreeing on common terminology is vital to support the development of EMS feedback.

All initiatives in our survey involved feedback from a hospital trust or regional network to EMS staff. Within the UK, the work of paramedics routinely involves referrals to community services or primary care physicians [43,44], but we have yet to identify any feedback initiatives spanning these boundaries. A UK-wide study on EMS non-conveyance [45] found high variability in conveyance rates, with a systematic review suggesting that feedback interventions may improve patient safety in this area [46]. Our previous qualitative work suggests that EMS professionals strongly desire feedback on non-conveyed patients [9], indicating a clear need for feedback intervention development in this area, particularly in light of current policies encouraging non-conveyance [44].

Our analysis provided insight into the theoretical basis of existing EMS feedback interventions and the need to strengthen theory in this field. Participants’ understanding of ‘underlying theory’ encompassed various factors, such as expert opinion, replicating other initiatives, clinical governance, education, job satisfaction and continued professional development. In the social sciences, a theory is defined as “an ordered set of assertions about a generic behaviour or structure assumed to hold throughout a significantly broad range of specific instances” [47] (p. 9). There are different levels of theory, i.e., programme theories (as developed in this study), mid-range theories (such as CP-FIT, which are restricted to a subset of a social phenomenon within a specific context) and grand theories (which seek to offer a comprehensive meta-narrative applicable to any context) [22,48]. Our participants’ examples of theories deviated from the traditional definition or levels of theory, suggesting a possible misunderstanding or lack of awareness of existing feedback theory. The absence of theoretical foundations in feedback initiatives has been critiqued in other healthcare settings, as it impedes progress in feedback intervention science [20,49].

We have previously used CP-FIT—a mid-range theory of clinical performance feedback [16]—to explain participants’ involvement with feedback at an abstract level [9]. However, the CP-FIT definition of mechanisms as “underlying explanations of how and why an intervention works” [16] (p. 2) did not seem to fully address mechanisms as interpreted within our chosen realist methodology, i.e., “a combination of resources offered by the social programme under study and stakeholders’ reasoning in response” [34] (p. 3). Adding an established list of mechanisms of action from behaviour change theory [39] to our data analysis allowed us to consider in-depth psychological processes by which outcomes were achieved. Utilising realist methodology allowed us to explore how these mechanisms linked to contexts and outcomes. This enabled the current study to go further than our systematic review of the published literature by adding depth of understanding [2].

### 4.1. Implications and Future Research

Our study demonstrates that feedback initiatives are common within EMS but that there is considerable divergence between research evidence into effective audit and feedback gleaned largely from other healthcare settings and implementation within EMS. Therefore, an important question is whether this implementation gap is due to unique challenges in the EMS setting for this type of intervention or other factors that can be addressed through future implementation research. Our findings suggest that the traditionally poor feedback culture within ambulance services and the practical difficulty of sharing data across hospital-EMS boundaries may be reasons for this divergence.

Our study demonstrates that existing theory can be used to evaluate existing feedback initiatives within EMS. Our programme theory will inform improvements and changes to ongoing EMS feedback initiatives. To support this, we are developing a best-practice guide for EMS feedback in collaboration with stakeholders based on our study findings.

Future research should explore EMS culture around the provision of feedback and seek to understand the prevalence, predictors and effects of feedback using quantitative methods. Our context–mechanism–outcome matrices (Table 3, Table 4 and Table 5) serve as a source of empirically based testable hypotheses for future research on EMS feedback. Our hypotheses expand others developed specifically within audit and feedback [12,14,16] to include ‘*pull*’ initiatives requiring feedback-seeking behaviour and alternative feedback types and outcomes, such as closure from patient outcome feedback. By not just limiting our review of practice to clinical performance feedback, we were able to develop a more holistic programme theory of feedback within EMS, allowing prospective initiative leads to consider a variety of feedback types and effects [50].

Although respondents anecdotally reported the effectiveness of local programmes, research is needed to provide robust quasi-experimental evidence to support feedback models in EMS. Our logic model expanded CP-FIT’s effectiveness outcomes [16] to include Proctor’s implementation outcomes [38] and could be used to design hybrid effectiveness-implementation trials of EMS feedback initiatives. Hybrid designs combine questions concerning intervention effectiveness with questions about how best to implement it in one study using a range of research designs [51,52].

### 4.2. Strengths and Limitations

Our review of current practice focused on EMS systems within the UK. Although a good representation of the 14 emergency ambulance services was achieved, it remains for future research to establish the degree to which these findings generalise to EMS in other health systems. The limits of existing international research in EMS feedback make generalisation difficult. However, our work draws upon established theory within the international implementation science audit and feedback domain, which should lay the groundwork for future comparative analysis. There is no prior literature regarding the number of EMS feedback initiatives in the UK to allow us to estimate the population size. However, we received many responses using a relatively small sampling framework, i.e., only eliciting responses from initiative leads.

Whilst our survey coverage of UK practice spanned 11 of 14 EMS, it did not include all EMS feedback initiatives currently or previously active. It was impossible to calculate a response rate for our survey, as we could not determine how many people received the survey due to the advertisement method. We received no responses from initiatives within London, South Central and the Isle of Wight, though the lead researcher’s professional network suggests the presence of feedback initiatives in some of those geographical areas.

The study protocol asked participating ambulance trusts to identify feedback initiatives based on or by identifying existing data-sharing agreements. However, we avoided asking initiative leads what these were. Some ambulance trusts expressed concern that not all initiatives would have robust information governance approvals. Initiatives without these approvals were, therefore, potentially underrepresented in our sample.

Our case study approach facilitated insightful analysis of specific practice initiatives that would not have been possible with other study designs. We collected less data for Case study 2 than other sites, which may have introduced selection bias. This was due to difficulty identifying participants who had received feedback despite extensive efforts to liaise directly with frontline ambulance staff, initiative leads and the ambulance service’s research department. We were unable to determine whether this was due to the feedback being so embedded in routine practice that recipients were unaware of it being linked to a formal feedback initiative, a lack of willingness to participate in research or disparity between actual practice and initiative leads’ perception of staff engagement with the initiative.

## 5. Conclusions

This study suggests that initiatives providing feedback for EMS staff are common in practice, with most UK ambulance trusts currently having examples of feedback initiatives within their footprint. EMS feedback initiatives are motivated by various factors but are challenging to implement and sustain effectively. The published literature differs from current practice in the focus and scope of EMS feedback initiatives. There is considerable opportunity to strengthen the methodological and theoretical basis for innovations in practice in this area. This includes providing guidance on the design of interventions and robust evaluation of different feedback approaches, especially those which might be unique in meeting the challenges of the EMS setting.

## Figures and Tables

**Figure 1 healthcare-11-02229-f001:**
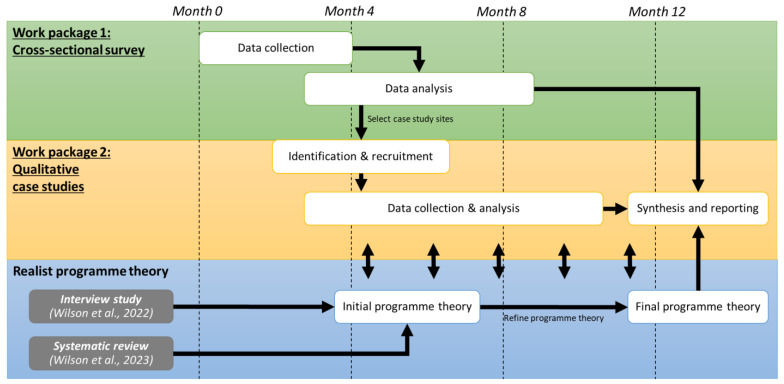
Study flow chart [2,9].

**Figure 2 healthcare-11-02229-f002:**
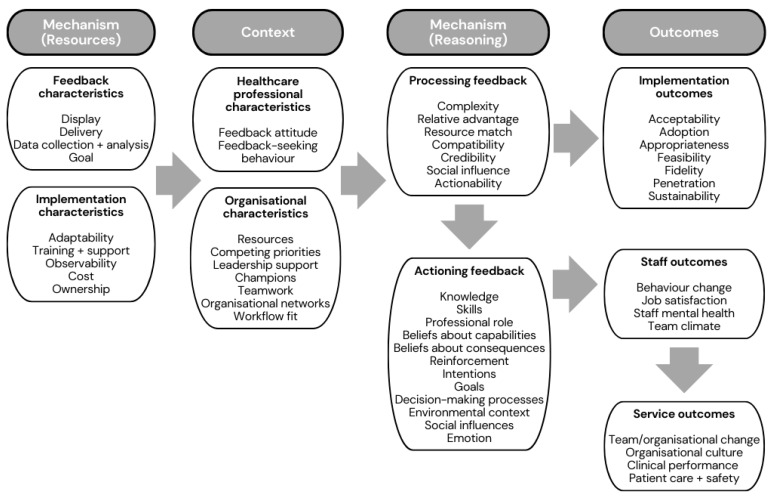
Logic model depicting our programme theory of EMS feedback.

**Table 1 healthcare-11-02229-t001:** Characteristics of feedback initiatives in EMS (n = 40).

Characteristics		N	%
**Feedback type**	Patient outcome feedback	31	77.5
	Clinical performance feedback	4	10.0
	Clinical performance + patient outcome feedback	5	12.5
**Currently active**	Yes	28	70.0
	No	12	30.0
**Year started** Median 2019 (IQR 2017–2020)	2004–2013	2	5.0
	2014–2018	13	32.5
	2019–2022	23	57.5
	No response	2	5.0
**Instances of feedback**	<10	1	2.5
	10–49	14	35.0
	50–99	11	27.5
	100–199	4	10.0
	200–299	2	5.0
	300–399	2	5.0
	400–499	1	2.5
	>500	2	5.0
	No response	3	7.5
**Feedback recipient** (categories are not mutually exclusive)	Frontline EMS staff	36	90.0
	Helicopter EMS staff	14	35.0
	EMS managers	9	22.5
	Emergency operations centre staff	2	5.0
	EMS organisation	3	7.5
	Combination	16	40.0
**Format**	Email or electronic document	35	87.5
	Face-to-face	3	7.5
	Written letter	1	2.5
	No response	1	2.5
**Frequency**	Daily	2	5.0
	Weekly	3	7.5
	Bi-weekly	1	2.5
	Monthly	3	7.5
	Quarterly	1	2.5
	Ad hoc	29	72.5
	No response	1	2.5
**Lag time** Median 21 days (IQR 7–30 days)	Within 24 h	3	7.5
	Within 7 days	8	20.0
	1–2 weeks	6	15.0
	2–4 weeks	14	35.0
	4–12 weeks	6	15.0
	>12 weeks	2	5.0
	No response	1	2.5
**Recipient level**	Individual	32	80.0
	Individual + team	3	7.5
	Individual + organisation	1	2.5
	Individual + team + organisation	4	10.0
**Patient cases**	Individual	38	95.0
	Aggregate	2	5.0
**Push or pull**	Push	8	20.0
	Pull	25	62.5
	Combination	6	15.0
	No response	1	2.5
**Stand-alone**	Stand-alone	34	85.0
	Part of an organisational or education initiative	6	15.0
**Action plan included**	Yes	12	30.0
	No	28	70.0
**Underlying theory**	Yes	15	37.5
	No	24	60.0
	No response	1	2.5
**Aim** (categories are not mutually exclusive)	Aid reflection, education + learning	21	52.5
	Provide feedback (“*close the loop*”)	19	47.5
	Improve staff wellbeing + job satisfaction	9	22.5
	Improve patient care	6	15.0
	Improve relationships between EMS + hospital staff	6	15.0
	Increase protocol adherence	4	10.0
	Satisfy EMS staff’s curiosity	3	7.5
	Clinical governance	2	5.0
**Barriers** (categories are not mutually exclusive)	Information governance	18	45.0
	Time and resources required to generate the feedback	16	40.0
	Technology	7	17.5
	No barriers	4	10.0
	EMS managers	4	10.0
	Lack of awareness from EMS staff that the initiative existed	3	7.5
	Identifying hospital collaborators and securing buy-in from hospital staff	2	5.0
	Concerns from EMS staff that errors identified in clinical practice would be used against them	2	5.0
**Facilitators** (categories are not mutually exclusive)	Collaboration	14	35.0
	Enthusiasm from EMS staff	9	22.5
	Engagement from hospital staff	8	20.0
	Initiative lead with time + enthusiasm	2	5.0
	Alignment with organisational priorities	2	5.0
	No existing initiative providing feedback	1	2.5
	Information governance	1	2.5
	Conducting a pilot of the initiative first	1	2.5
	Inclusion of specific expertise within the team (e.g., data analyst)	1	2.5

**Table 2 healthcare-11-02229-t002:** Characteristics of case study sites.

Characteristics	Case Study 1	Case Study 2	Case Study 3	Case Study 4
**Data sources**	Initiative lead interview (n = 1), Feedback recipient interviews (n = 3), documents (n = 2)	Initiative lead interviews (n = 2), Feedback recipient interview (n = 1), documents (n = 1)	Initiative lead interviews (n = 2), Feedback recipient interviews (n = 3), documents (n = 3)	Initiative lead interview (n = 1), Feedback recipient interviews (n = 4)
**Context**	Rural	Urban	Rural	Rural, urban + suburban
**Catchment area**	1 hospital	Regional	2 hospitals	National
**Initiative lead**	EMS manager	Regional network clinician + EMS specialist clinician	EMS specialist clinician + EMS manager	EMS specialist clinician
**Patient group**	All patients	1 condition	All patients	1 condition
**Feedback type**	Patient outcome	Clinical performance	Patient outcome	Clinical performance + patient outcome feedback
**Push or pull**	Pull	Push	Pull	Push
**Year started**	2018	2015	2014	2020
**Instances of feedback**	50–99	Unknown	>500	10–49
**Feedback recipient**	Frontline EMS staff	Frontline EMS staff	Frontline EMS staff	Frontline + helicopter EMS staff
**Format**	Email	Face-to-face	Electronic dashboard	Email
**Frequency**	Whenever requested	When breach occurred	Weekly	After every event
**Lag-time**	2–4 weeks	A few weeks	Up to 7 days	1–6 months

**Table 3 healthcare-11-02229-t003:** Context–mechanism–outcome matrix of the implementation of EMS feedback interventions.

Mechanism (Resources)		Context		Outcomes		Mechanism (Reasoning)		Data Sources
*If*	an initiative is co-designed	*with*	feedback recipients that have a positive feedback attitude and leadership support	*then*	acceptability is likely to be higher ↑	*because*	there is a sense of ownership, understanding and credibility	CS1-P1, CS1-P3
*If*	a senior clinician provides push feedback on protocol adherence	*within*	an organisation that has leadership support	*then*	it is likely to be accepted ↑	*because*	there is a sense of credibility	CS2-P1, CS2-P2, CS4-P2, CS4-P3
*If*	an initiative provides patient outcome feedback via a ‘pull’ initiative	*to*	healthcare professionals that display feedback-seeking behaviour	*then*	adoption and appropriateness are likely to be higher ↑	*because*	there is a sense of compatibility	CS1-P2, CS1-P3, CS1-P4, CS3-P2, CS3-P3
*If*	a feedback initiative provides patient outcome feedback	*within*	an organisation where informal follow-up opportunities are limited	*then*	adoption is likely to be higher ↑	*because*	there is a relative advantage	CS1-P1, CS1-P1, CS3-P2, CS3-P3
*If*	the initiative lead guides feedback providers on paramedics’ scope of practice	*when*	feedback providers do not have any experience or knowledge of EMS clinical practice	*then*	feedback is more likely to be appropriate ↑	*because*	it is actionable	CS1-P1, CS4-P1
*If*	feedback is meant to be generated with limited training	*within*	organisations that can appoint champions to generate feedback	*then*	it is more likely to be feasible ↑	*because*	there is a resource match	CS1-P1, CS2-P1
*If*	the number of processed feedback requests is limited by having specific eligibility criteria	*within*	an organisation that has limited resources ring-fenced for this initiative	*then*	sustainability would be better ↑	*because*	there would be a better resource match	CS1-P1, CS4-P1
*If*	patient outcome feedback does not identify which patient it relates to	*within*	an organisation where staff attend multiple patients per shift and feedback is delayed	*then*	acceptability and adoption are likely to be poor ↓	*because*	it would be complex for recipients to identify whom the feedback related to	CS1-P4 CS3-P5, CS4-P1, CS4-P2
*If*	feedback is pushed out without training, support or raising awareness	*within*	an organisation that traditionally has poor feedback culture	*then*	acceptability is likely to be poor ↓	*because*	there is poor compatibility	CS2-P3, CS4-P2, CS4-P3, CS4-P4
*If*	feedback is provided with a long lag-time	*to*	a staff member that can informally follow up on patients	*then*	adoption is likely to be poor ↓	*because*	there is no relative advantage	CS4-P1, CS4-P2, CS4-P3, CS4-P4
*If*	an ambulance trust leads a feedback initiative	*without*	clearly articulating how the feedback is going to be used by the trust	*then*	adoption may be low ↓	*because*	it could be perceived as being punitive	CS1-P2, CS1-P4, CS4-P2, CS4-P4
*If*	a feedback initiative has a high resource cost for one individual	*within*	an organisation where there are competing priorities	*then*	it is not likely to be sustainable ↓	*because*	there is no resource match	CS1-P1, CS1-D2, CS4-P1

Note. ↑ indicates a positive outcome and ↓ indicates a negative outcome.

**Table 4 healthcare-11-02229-t004:** Context–mechanism–outcome matrix of interventions providing patient outcome feedback for EMS staff.

Mechanism (Resources)		Context		Outcomes		Mechanism (Reasoning)		Data Sources
*If*	patient outcome feedback allows comparisons between prehospital working impression and hospital diagnosis	*in*	an organisation that monitors this on an aggregate level	*then*	it is likely to improve patient safety ↑	*because*	Knowledge is improved by organisations arranging additional training; the environment is changed by organisations increasing alternative pathways	CS3-D1, CS3-D2, CS3-D3, CS3-P3, CS4-P1
*If*	patient outcome feedback is provided	*to*	ambulance staff interacting with patients who are anxious about what might happen at the hospital	*then*	it is likely to improve service quality ↑	*because*	staff have a belief about consequences and can reassure patients	CS1-P2, CS3-P3
*If*	patient outcome feedback is provided regarding patients with a difference in prehospital/hospital diagnosis, patients frequently calling the ambulance service or patients with common conditions	*to*	healthcare professionals that have a positive feedback attitude and within an organisation that supports autonomous decisions by staff	*then*	it is likely to lead to individual behaviour change and better patient care ↑	*because*	it improves beliefs about consequences	CS1-P1, CS1-D1, CS3-P3, CS3-P5, CS4-P2, CS4-P5
*If*	additional learning materials are provided alongside patient outcome feedback	*to*	healthcare professionals that have a positive feedback attitude	*then*	it is likely to lead to individual behaviour change and better patient care ↑	*because*	it improves knowledge	CS4-P2, CS1-P1, CS3-P5
*If*	an initiative provides patient outcome feedback via a ‘pull’ initiative	*to*	healthcare professionals that display feedback-seeking behaviour	*then*	it may lead to behaviour change and improved clinical performance/patient safety ↑	*because*	it improves knowledge and decision-making processes	CS1-P1, CS1-P2, CS1-P3, CS3-P2
*If*	patient outcome feedback is provided on patients whom staff have a particular interest in	*to*	staff who do not routinely find out what happens to their patients	*then*	it is likely to improve staff mental health ↑	*because*	it allows an element of closure	CS1-P1, CS1-P2, CS1-P4, CS1-D1, CS3-D2
*If*	a senior clinician sets up an initiative providing patient outcome feedback	*within*	an organisation that has traditionally viewed patient outcome information as an ‘add on’	*then*	it is likely to lead to increased job satisfaction ↑	*because*	it makes staff feel appreciated	CS3-P3, CS3-P4, CS4-P2, CS4-P3, CS4-P4, CS4-P5
*If*	patient outcome feedback is provided only by emergency department staff	*but*	patients are admitted to hospital wards	*then*	staff behaviour change may be limited ↓	*because*	there is only a partial understanding of the consequences	CS1-P1, CS1-P3, CS3-P5

Note. ↑ indicates a positive outcome and ↓ indicates a negative outcome.

**Table 5 healthcare-11-02229-t005:** Context–mechanism–outcome matrix of interventions providing clinical performance feedback for EMS staff.

Mechanism (Resources)		Context		Outcomes		Mechanism (Reasoning)		Data Sources
*If*	pathway adherence feedback is provided	*within*	an organisation that has a centralised model of care (i.e., bypassing local hospitals to attend specialist treatment centres)	*then*	it will lead to behaviour change and improved patient safety ↑	*because*	staff will have had the pathway reinforced	CS2-P1, CS2-P2, CS4-P2, CS4-P5
*If*	push-feedback on protocol adherence is provided	*to*	a healthcare professional that has a positive feedback attitude	*then*	it is likely to lead to behaviour change ↑	*because*	it improves knowledge	CS2-P1, CS2-P2, CS3-P2, CS3-P3, CS4-P2, CS4-P4
*If*	feedback on protocol adherence is provided	*within*	an organisation that prioritises the protocol topic	*then*	it is likely to improve clinical performance ↑	*because*	it improves knowledge and skills	CS2-P1, CS2-P2, CS4-P1
*If*	push-feedback is provided	*to*	a healthcare professional that has a low sense of confidence	*then*	it may negatively affect staff mental health ↓	*because*	it reduces beliefs about capabilities	CS1-P3, CS2-P1, CS4-P2, CS4-P4

Note. ↑ indicates a positive outcome and ↓ indicates a negative outcome.

## Data Availability

The datasets generated and analysed during the current study are not publicly available as sharing the raw data would violate the agreement to which participants consented; however, the manuscript includes an extensive list of quotes, and the datasets are available from the corresponding author on reasonable request.

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
