# Peer review of "Feedback for Emergency Ambulance Staff: A National Review of Current Practice Informed by Realist Evaluation Methodology"

_healthcare, 2023, doi:10.3390/healthcare11162229_

Round 1
Reviewer 1 Report
The study provides an overview of feedback systems used in UK ambulance services and sets out a programme theory of how these feedback systems work. This is an importat issue, and the study provides significant new information and the theory presented structures the phenomenon well. EMS needs such non-clinical research to evolve.
However, there are few issues I would like to point out:
#1 In introduction, receiving feedback in EMS setting is described as "vital", based on authors' previous research, but in broader scope the effect of feedback is reported to vary (p. 2, 52-61). What could be the reason for this?
#2 The study utilizes mixed methods, which is reasonable in view of the research questions. Figure 1 describes the research process well. In case study interviews reported in Table 2, it seems that number of interviewed initiative lead and feedback recipients vary from 1 to 4. Especially in "pull feedback" system the experience and perception of feedback recipient may affect the findings. Did you consider to get few more paramedics to be interviewed for each studied case to ensure to find all relevant aspects? You have dealt with the issue in discussion (p. 17 504-511) but perhaps you should discuss further if this may cause some kind of selection bias or is it just irrelevant to the results?
#3 You refer feedbacks as "reports" throughout the manuscript. For a reader like me who is not deeply familiar with EMS in UK, "report" does not accuraely describe its meaning. Does it refer to an individual feedback for a paramedic involved to the patient care, or some kind of summary of individual feedbacks? Most initiatives have generated less than 100 reports, but it is not clearly stated over what period of time these reports were generated. If possible, I would also like to see some reference how common it is to receive or ask such feedback: how many times per a paramedic a year, or missions per a feedback would be interesting figures. Perhaps you could also consider add a sample raport as a supplementary file?
#4 Table 3 would be perhaps easier to assimilate if it were grouped or ordered by outcome, or the direction of each outcome would be indicated by +/- or up/down arrows.
Despite these minor notes and observations, the study provides further understanding and structure to the operation of EMS feedback systems. Thanks!
Author Response
Dear Reviewer 1, thank you for your positive comments concerning the importance of this topic area and our chosen approach to this study. Below we respond to each of your comments in turn.
#1 - Feedback is both 'vital' and its 'effects vary', these can both be true at the same time for EMS and healthcare more broadly. To make this more clear we have reiterated on lines 52-53 that 'while feedback overall has small to moderate positive effects, these effects vary'.
#2 - Thank you for correctly pointing out that the number of interviews with feedback recipients varied across the case studies (CS1 n=3, CS2 n=1, CS3 n=3, CS4 n=4). We agree that capturing the experience of feedback recipients was vital to the research question and did attempt to recruit more participants for CS2, where only 1 feedback recipient was interviewed. However, as we reflect upon in the discussion section (lines 504-512), we were unsuccessful with our attempts. In an effort to be even more transparent we have now added "which have introduced selection bias" (line 511).
#3 - We have now changed 'reports' to "instances of feedback" to convey that this is relating to individual instances of feedback. Unfortunately, this study did not seek to answer how frequent feedback is so we are unable to provide this information. However, we agree that such a study is important to undertake and have added "prevalence" to line 470 where we talk about future research that is needed. We are also unable to provide an sample feedback report as a reference, as case study sites did not share this with us, nor would our ethical permissions allow us to disseminate this further if we were to have received this from case study sites.
#4 - We have added up/down arrows to Table 3 and reordered the outcomes to show to 'positive' outcomes first, followed by 'negative' outcomes.
Reviewer 2 Report
This is a paper purporting to show how ambulance services in the UK interact with various forms of feedback they receive in order to deliver better care.
There is some interesting data about the various forms of feedback these ambulance services receive. There should definitely be more feedback regarding patient outcomes and this is a major research finding.
The quality of writing is excellent. I would recommend this article for publication after minor changes:
Development of programme theory : although there is a short description of the programme theory, it could be described in more detail for those not familiar with realist evaluation, especially the articulation between Context Mechanism and Outcome.
I would encourage the authors to follow-up with a quantitative study to ascertain some of the CMO findings with more data, this could be a perspective for the study.
Author Response
Dear Reviewer 2,
Thank you for your kind comments regarding our interesting topic choice and excellent writing quality. In response to your comments we have:
#1 - Added further detail regarding the programme theory development and particularly the articulation between context, mechanisms and outcomes: "These configurations set out the causal links between the specific mechanisms that trigger intervention outcomes within particular contexts [31,32]. Context here pertains to the background and implementation setting, while mechanisms describe how resources bring about change and outcomes relate to intended and unintended consequences [32-34]."
#2 - Added in "using quantitative methods" in line 470+471 when talking about future research that is needed before our sentence highlighting that "Our context-mechanism-outcome matrices (Tables 3-5) serve as a source of empirically-based testable hypotheses for future research on EMS feedback."
Reviewer 3 Report
The abstract outlines a study conducted to understand how feedback is provided to Emergency Medical Service (EMS) professionals in the United Kingdom (UK) and develop a program theory of how feedback works within EMS. The study used a mixed-methods, realist evaluation framework to explore the mechanisms by which feedback affects EMS professionals' quality of care and professional development.
Key Points from the Abstract:
-
Feedback in EMS: Research suggests that feedback in EMS positively impacts the quality of care and professional development of EMS professionals. However, the specific mechanisms by which feedback achieves these effects need further exploration.
-
Study Aim: The study aimed to understand how UK ambulance services provide feedback to EMS professionals and develop a program theory of how feedback operates within EMS.
-
Methodology: The study used a mixed-methods approach, incorporating both quantitative and qualitative methods, and followed realist evaluation methodology to uncover the underlying principles of how feedback influences outcomes within the EMS context.
-
Data Collection: The study involved a national cross-sectional survey to identify feedback initiatives in UK ambulance services, followed by four in-depth case studies with qualitative interviews and documentary analysis.
-
Feedback Initiatives: The survey revealed that feedback initiatives in UK ambulance services mainly provided individual patient outcome feedback through "pull" initiatives triggered by staff requests.
-
Challenges: Challenges related to information governance were identified in the context of providing feedback.
-
Program Theory: The study developed a program theory of feedback to EMS professionals, encompassing context (healthcare professional and organizational characteristics), mechanisms (feedback and implementation characteristics, psychological reasoning), and outcomes (implementation, staff, and service outcomes).
-
Implications: The study suggests that most UK ambulance services use a range of feedback initiatives, and it provides empirically-based testable hypotheses for future research.
-
Clinical Performance Feedback: The abstract also highlights that feedback on clinical performance is well-researched and can result in small to moderate improvements in patient care across various healthcare settings, including EMS.
Overall, the study aims to shed light on how feedback is currently provided in the UK EMS setting and offers insights into the mechanisms through which feedback influences EMS professionals' performance and outcomes.
The article appears to be written in a formal and academic style. The language is generally clear and well-structured, and the author uses appropriate terminology related to the field of Emergency Medical Service (EMS) feedback. The sentences are coherent, and the ideas are well-expressed.
There are, however, a few areas where improvements could be made to enhance clarity:
-
Some sentences are quite lengthy and could be broken down into shorter ones to improve readability.
-
In a few instances, the wording could be simplified to make it more accessible to a broader audience.
-
The use of acronyms (e.g., EMS, CP-FIT) is prevalent, and while they are explained initially, it might be helpful to reiterate their meanings occasionally to aid reader comprehension.
Overall, the quality of English language in the article is good, but some minor adjustments could make it even clearer and more approachable for a wider audience.
Author Response
Dear Reviewer 3,
Thank you for reviewing our manuscript. In response to your comments, we have made the following changes:
#1 - We have shortened several of the sentences throughout the manuscript, for example line 44, 445, 453 and 526.
#2 - We have simplified the wording in the manuscript, for example replacing 'diverges' with "differs" and 'hampers' with "impedes".
#3 - We have re-iterated the meaning of less familiar abbreviations (CMOCs, CP-FIT), e.g. line 444+445 "CP-FIT - a mid-range theory of clinical performance feedback".